# Custom-Made 3D Titanium Plate for Mandibular Reconstruction in Surgery of Ameloblastoma: A Novel Case Report

Somangshu Chakraborty [1,*] , Rajdeep P. Guha [2] , Sukanya Naskar [2] and Rajarshi Banerjee [1]

1   Haldia Institute of Dental Sciences and Research, Haldia 721645, India
2   Chittaranjan National Cancer Institute, Kolkata 700156, India
*   Correspondence: somichax94@gmail.com

**Abstract:** Ameloblastoma is a benign yet locally invasive odontogenic neoplasm, characterised by slow growth and painless swelling. The treatment for ameloblastoma varies from curettage to en bloc resection, with recurrence commonly occurring. The safety margin of resection is hence essential to avoid recurrence. Understanding the three-dimensional anatomy for reconstruction of mandibular defects after tumour resection often poses problems for head and neck surgeons. Historically, various autografts and alloplastic materials have been used in the reconstruction of these types of defects. Over time, advances in technology with computed tomography scanners and three-dimensional images enhance the surgical planning and management of maxillofacial tumours. The development of new prototyping systems provides accurate 3D biomodels on which surgery can be simulated, especially in cases of ameloblastoma, in which the safety margin is vital for the clinical outcome. The objective of this paper was to report a clinical case of employing these methodologies for reconstruction after an extensive mandibular resection. The clinical outcomes were observed. A case of follicular ameloblastoma of the mandible is depicted in the following paper, where a 3D biomodel was used throughout the surgery. A 3D printed patient-specific titanium implant was manufactured and placed intraoperatively for reconstruction. The treatment had satisfactory postoperative results without complications. Titanium implants being bioinert, customisable and easily workable, especially with the help of 3D virtual planning techniques, can be considered as ideal alloplastic materials for mandibular reconstruction.

**Keywords:** ameloblastoma; stereolithography; custom-made three-dimensional titanium implant; mandibular reconstruction

## 1. Introduction

Ameloblastoma is a clinically significant odontogenic neoplasm. It is a locally invasive polymorphic tumour having follicular or plexiform patterns in a fibrous stroma which is anatomically benign [1]. Ameloblastoma occurs more in the mandible (80%) and less in the maxilla (20%). In the lower jaw, 70% of ameloblastomas are located in the molar or ascending ramus region [2]. It shows a peak of incidence in the fourth decade of life without any gender predilection. Ameloblastomas are categorised into four types: unicystic, solid or multicystic, peripheral and malignant. A well-circumscribed unilocular/multilocular radiolucency is exhibited radiographically [3].

Ameloblastoma is mostly asymptomatic and is noticed incidentally in orthopantomography. The common symptoms are facial swelling, pain, malocclusion, loose teeth, ill-fitting dentures, periodontal diseases, oroantral fistulas and airway obstruction [4].

Depending on treatment, ameloblastoma recurrence rates from 15% to 90% are reported [5]. Therefore, wide local resection of the jaw comprising margins of 1 cm or more is preferred.

The literature depicts both conservative approaches and radical procedures as management. While smaller lesions are generally treated by a less aggressive modality, larger lesions require a radical surgical tumour ablation, resulting in large defects and making reconstruction difficult [6]. Treatment for ameloblastoma varies from curettage to en bloc resection. Reconstruction of the affected area is necessary after the resection. Bone-free flaps are the standard treatment approach for the reconstruction of significant mandibular defects; thus, the main bone sources include the fibula, scapula and iliac crest [7]. Although these methods offer the filling of required tissue and support, distinct limitations have been described in the literature, in other words: the establishment of a new surgical place, plate exposure or fractures, complications with articulation and several aesthetic consequences [8]. Taking into account these aspects mentioned, the objective of this paper is to report a clinical case of ameloblastoma of the mandible in a patient where a 3D implant was chosen for suitable reconstruction.

Recently, in patients having refusals or contraindications to free flaps, 3D titanium plates have been used. This paper reports a case of follicular ameloblastoma in the mandible submitted to surgery, using a selective laser sintering biomodel for presurgical planning, utilising 3D biomodels for reconstruction with a custom-made porous titanium plate. Revolutionary imaging techniques and the development of new rapid prototyping systems such as selective laser sintering and stereolithography impact on achieving exact three-dimensional (3D) images. These technologies can manufacture physical models from Computer Aided Design (CAD) by means of 3D printers. A printed 3D model can assist in the fabrication of patient-specific implants, contouring plates and/or planning of bone graft harvest geometry before surgery [9]. Stereolithography and 3D printing are two frequently used rapid prototyping technologies. In stereolithography, a liquid resin is polymerised by laser forming the desired solid material. Selective laser sintering creates models from heat fusible powders, such as polycarbonate and glass-filled composite nylon, by tracing a modulated laser beam across a solid thin slice. Heating causes the particles to fuse together, creating a solid layer that is again covered by powder, and the next slice is formed on top, eventually completing the object. This technique, more commonly used, has ample applications in craniofacial surgery.

## 2. Case Report

A 41-year-old female patient, a household maid by occupation, reported to the Department of Head and Neck Oncology at a tertiary cancer centre in Kolkata, India, complaining of a swelling in the lower third of her face for almost six years. Her mandible appeared with a gross deformity owing to the swelling. There was no pain associated with the swelling. The patient did not have any relevant medical, family or psychosocial history attributing to her disease. A solitary ill-defined diffuse swelling was observed over her central chin and right mandibular body regions, measuring approximately $8 \times 14$ cm in dimensions. Anteroposteriorly, this swelling extended from the parasymphysis region of the left cheek to the angle region of the mandible of the right side and, superioinferiorly, from the pretragal region to the lower border of the mandible of the right side (Figure 1A). The pathology was insidious in onset and gradually increased to the presenting size. There was no history of previous medical intervention for this case. There was no history of trauma or toothache or discharge from the swelling over the years. Intraoral clinical examination revealed both buccal and lingual cortical expansions and partial edentulism in the lower alveolus. The swelling had obliterated the vestibular depths and involved the floor of the mouth elevating the tongue position (Figure 1B). The overlying mucosa appeared stretched with a loss of keratinisation and impaired vascularity. Palpation showed buccal cortex erosion with "egg-shell" appearance in the right posterior mandible. The overlying skin from outside appeared stretched with normal temperature. The lesion had no intraoral communication. In radiographic evaluation, a CT scan of the face showed a large lytic expansile multiloculated lesion involving the symphysis menti and the alveolar process on

the right side of the body of mandible (Figure 2). Soft tissue punch biopsy from the anterior mandible confirmed a follicular type of ameloblastoma for this patient.

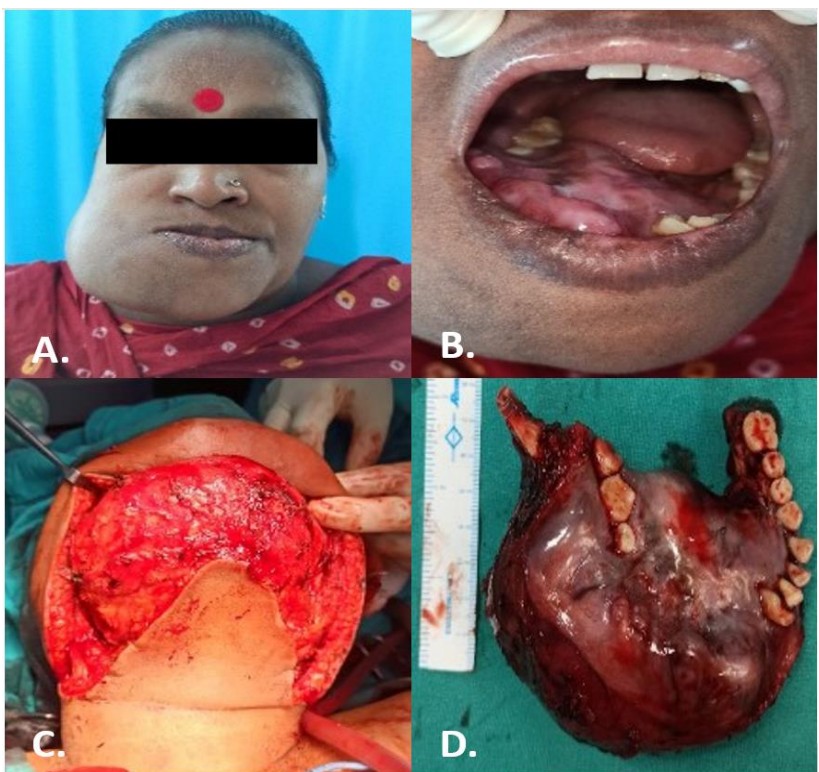

**Figure 1.** Preoperative records and resection. (**A**) Preoperative profile of patient, (**B**) Intraoral lesion, (**C**) Ameloblastoma, (**D**) Composite specimen.

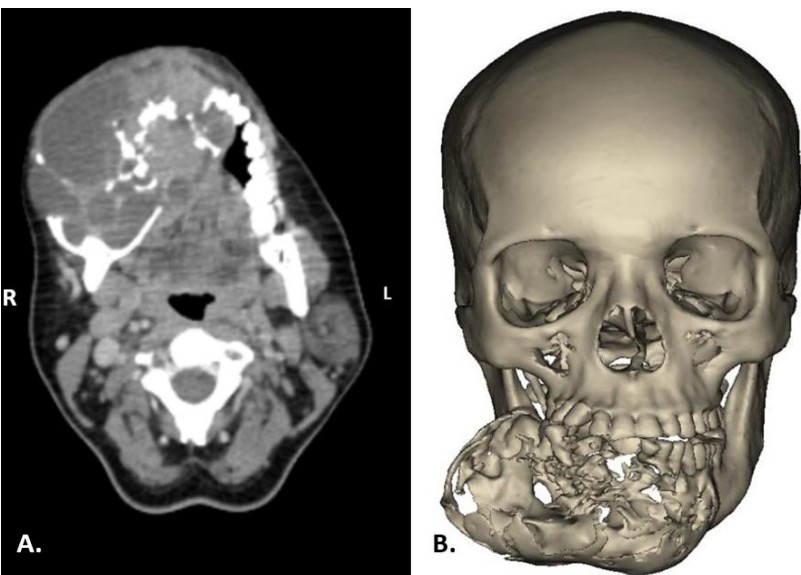

**Figure 2.** Preoperative CT scans. (**A**) Axial CT window showing the extent of the disease and bone involvement, (**B**) Virtual 3D CT frontal view.

For surgical intervention, a mandibular resection in the healthy areas was chosen based on a 3D CECT model, and those margins were provided with the help of specific prefabricated cutting guides. Reconstruction was then undertaken utilising a 3D custom-

made porous titanium plate. The written consent of the patient according to ethical principles was signed.

Once anaesthesia was achieved via naso-endotracheal intubation, "visor" [10] incisional approach was made extraorally. The entire mandibular body with the tumour was exposed after dissection through anatomic planes (Figure 1C). Intraoral margins were kept and incised for en bloc resection of the pathology (Figure 1D). Slim tomodensitometric sections helped create 3D rehabilitation. The cutting guides and custom-created 3D plate were fabricated employing a certain medical software via computer-aided design and manufacturing (CAD-CAM) (Figure 3D). The cutting guides were fixed considering the formerly described markers at the level of the right mandibular ramus and the left mandibular body area (Figure 3E). In the titanium implant, no coating was provided owing to the implant manufactured by the fusion of titanium powder having an inherent rough surface through 3D printing. The implant served as a reconstruction plate and had a porous body; hence, soft tissue integration around the implant was predictable (Figure 3F). The plate was subsequently retained in position using screws for fixation. Non-locking bicortical screws of profile 2 mm and 8 mm length were used, with a total of four screws at each end of the plate fixed to achieve stability and load-bearing properties. Once the implant was fixed, a predictable tension in soft tissue around it was observed and a soft tissue void intraorally became a probability (Figure 4A). Therefore, a submental flap was designed and placed to replenish the missing floor of the mouth (Figure 4C). A tracheostomy was performed to establish the airway. The patient was educated on postoperative oral hygiene guidelines and possible difficulties associated with the prosthesis. She was kept on naso-gastric Ryle's tube feeding postoperatively. The patient was recalled after 1 month and was trained to start with oral feeding and the tracheostomy tube was removed. The postoperative results were found to be satisfactory (Figure 4D).

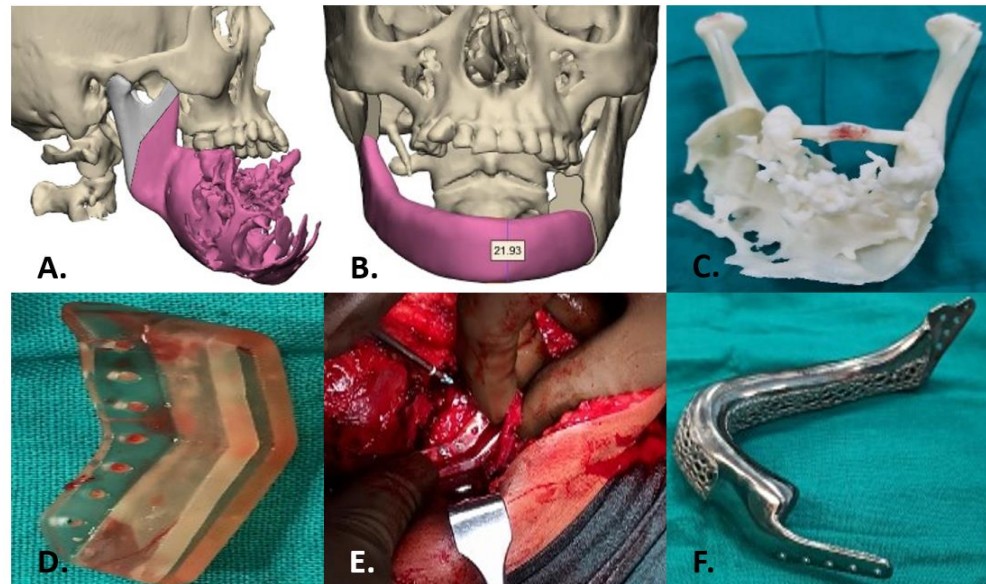

**Figure 3.** Virtual planning, biomodel and implant. (**A**) Scan with margin markings, (**B**) Planning of patient-specific implant, (**C**) Stereolithography 3D model, (**D**) Cutting guide, (**E**) Guides used intraoperatively prior to resection, (**F**) Custom-made 3D porous titanium implant.

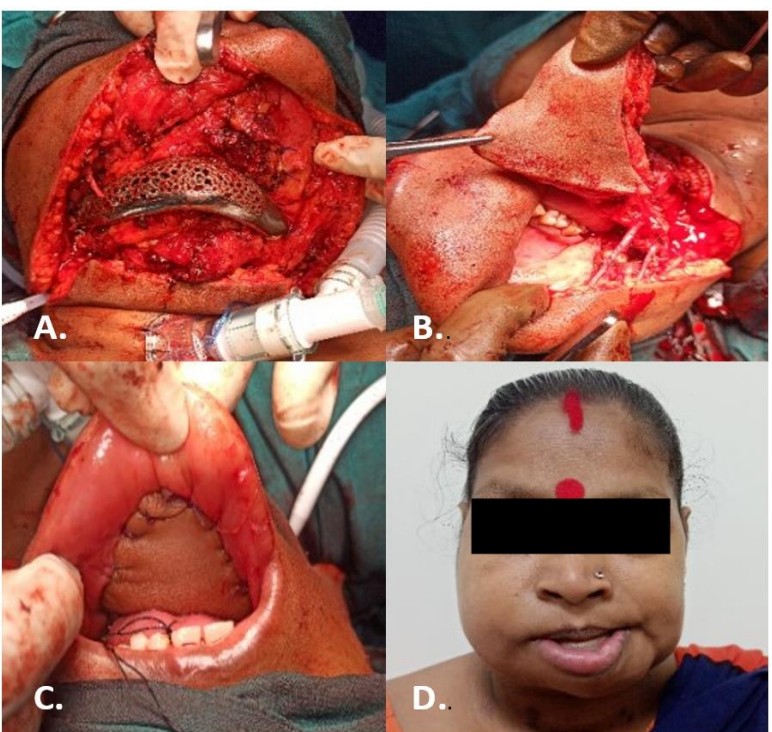

**Figure 4.** Reconstruction and follow-up. (**A**) Placed implant, (**B**) Submental flap, (**C**) Intraoral closure, (**D**) One-month postoperative profile.

### 3. Discussion

Due to financial constraints and the essentially asymptomatic nature of the swelling, this native from the eastern part of India could not afford medical treatment for a lengthy period of time until people known to her brought her to our institution. Examining the extent of her disease and other systemic factors, we opted for 3D reconstruction and planned our surgical work-up.

The 3D scheme permits the preservation of anatomic subtlety and the application of cutting guides inevitably shortens the surgical time. Impeccable rearrangement of the inferior border of the mandible and reinforced reconstruction of mandibular shape and strength are obtained in contrast to traditional mega-plates [11]. Cohen et al. stated that 3D printing technology provides a precise and fast mandibular reconstruction, which aids in a shortened operation time (therefore, decreased exposure time to general anaesthesia, decreased blood loss and shorter wound exposure time) and an easier surgical procedure [12]. Moreover, the porous titanium permits tissue ingrowth from the adjacent tissue, allowing improved integration [13]. Some authors have advocated that the mechanical strength of these 3D plates is equal to that of cortical bone [14]. In the case of large tumours, radical surgery prevents relapse of the tumour on a long-term basis. Three-dimensional reconstruction based on tomography scans enhances visualisation of the complexity of the defect after ameloblastoma resection. The goal is to return to premorbid shape and function.

However, one probable demerit of 3D custom-made titanium plates is that the long-term tolerance of these implants has not been recognised, but it can be assumed that the tolerance is undoubtedly superior to that of traditional plates [15]. Even though 3D CAD-CAM porous titanium was used in this case, the possibility of alloplast material exposure remains. These customised implants are generally expensive and patients should be guided accordingly.

Sykes et al. evaluated the accuracy of the prototype with conventional methods for reproduction in maxillofacial prostheses, duplicating the non-affected anatomic structures by the prototyping method and comparing them with those obtained by silicone impression.

According to the authors, the main advantage of rapid prototyping is the high accuracy of prostheses obtained by this method [16].

Together with the cases previously described by Qassemyar et al. and Touré and Gouet [17], the different advantages presented by this type of 3D custom-made porous titanium plates for complete reconstruction are significant and treatment outcomes can be largely influenced.

## 4. Conclusions

Our experience of this case was vivid. With a thorough work-up and productive discussion on virtual planning, the surgery of ameloblastoma with a 3D reconstruction plate could be well executed. We concluded that 3D printing satisfactorily facilitated anatomic restoration and bone cutting guides significantly eased the accurate fixation of the implant. We would like to conclude that reconstructions by 3D custom-made porous titanium plates are simple, rapid and predictable. Titanium implants being bioinert, customisable and workable, especially along with 3D virtual planning techniques, can be considered as ideal alloplastic materials for mandibular reconstruction.

**Author Contributions:** Conceptualization, R.P.G.; methodology, R.P.G.; writing—original draft preparation, S.C.; writing—review and editing, R.P.G. and S.N.; supervision, R.P.G. and R.B. All authors have read and agreed to the published version of the manuscript.

**Funding:** This research received no external funding.

**Informed Consent Statement:** Written informed consent to publish this paper has been obtained from the patient.

**Data Availability Statement:** Not applicable.

**Conflicts of Interest:** The authors declare no conflict of interest.

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
