# Peer review of "Custom-Made 3D Titanium Plate for Mandibular Reconstruction in Surgery of Ameloblastoma: A Novel Case Report"

_2038-9582, doi:10.3390/std11030009_

Round 1

Reviewer 1 Report

Line 80: Would you please add a radiological image such as MRI or Xray 

Line 82: which pathological method did you prefer for diagnostic purposes at the first stage?

Line 84: Which coating method was used for the titanium implant to achieve osteointegration?

Line 85: Would you add a reference for ''visor'' approach?

Line 91: Which type ( locking ?) and size of screws was used?  How did you ensure os stabilitisation?  An intramedullary component of of implant would be better to achive better stability and weght sharing to the healty mandiular bone. 

Line 93: 1 month f follow -up very short period to evaluate outcomes. when did you let patient to eat and chew foods. How did you manage the 1st postoperative week? 

Line 96: Can you add a postoperative x ray ? 

Line 105 : Discussion is too short. Please add more studied about 3D custom made implants. 

Regards.

Author Response

Ma'am/Sir,
Thank you for your reviews.
I have added a figure citing the preoperative Ct scan axial view and the 3D Ct on virtual Window.
I have only these imaging datas. I will be obliged if you consider these and approve.
I have mentioned which pathological method was used for initial diagnosis of the disease.
Regarding the coating of the implant, I discussed and enquired the manufacturers of the implant, from where I got to know no coating was applied on this implant owing to its 3d production from heat fusible powdered titanium and a soft tissue ingrowth was expected into the porous plate.
In relation to the stability and fixation of the plate I have added a note on the types of screws used. we used 2 screws fixed intramedullary and 2 screws fixed at the lower border of mandible for stability and load bearing properties.
For the 1st post operative week, the patient was kept in I.C.U and I have added a note of the tracheostomy that was done for airway and nasogastric feeding that was done for the patient. Till one month post operatively she was on these tubes complied well. Then we removed them. The patient is still on follow up. Its been 3 months after the surgery and there are no complications at all as of now.
I sincerely regret to inform, we dont have a post operative x-ray for this patient. Kindly consider.
I have added another study and a few more points to enrich the discussion.
I could not address all your queries owing to certain limitations of our setup and patient profile. I indeed tried to input as much as I could. I sincerely regret if the manuscript falls short in some aspects. I would request you to approve and kindly consider it for publication.
Regards.

Reviewer 2 Report

To authors:

I congratulate and encourage the authors to finish this interesting manuscript with the requested suggestions to be corrected in the text.

1. Abstract

In the abstract I suggest reinforcing the objective of the manuscript, i.e. “The objective of this paper was to report a clinical case of…..”

2. Introduction

The introduction is well updated and well linked to the theme and basic objectives of the work.

What is unique about this case? What does it add new to the medical literature?

Again, I suggest inserting a clear objective at the end of the introduction. For example, “Taking into account these aspects mentioned, the objective of this paper was to report a clinical case of….”

3. Case report

Ethical aspects should be considered in this beginning of clinical description observing the authorization and consent of the patient for the planned treatment.

I suggest inserting in the text the indications of the figures, for example,

“Anteroposteriorly it extended from the parasymphysis region of left cheek to the mandibular angle of right side and superio-inferiorly from pretragal region to lower border of mandible in the right side (Figure 1A).”

 “Radiographs showed a large well-defined radiolucent mass with diffuse locular arrangements (Figure ???). It would be interesting to insert the radiographic exam.

Information that can improve and enrich the scientific clinical description of this case.

A. Demographic information (such as age, gender, ethnicity, occupation).

B. The patient's main symptoms (their main complaints).

C. Medical, family, and psychosocial history including comorbidities and relevant genetic information.

D. Previous therapeutic interventions and results

E. Adverse and unforeseen events during the surgical procedure.

4. Discussion

Discuss the clinical history of this patient. Has the patient remained without medical care until this moment? How can the patient allow this lesion to evolve, limiting her health and routine to this form? Has she had no previous treatment?

I suggest reinforcing the discussion with more information about the limitations of the use of this technique and the difficulties in its execution.

Discussion of strengths and limitations in case management

5. Conclusion:

I suggest inserting a conclusion item at the end of the discussion.

What are the main lessons learned from this case?.

Author Response

To reviewer 3,
Thank you for your reviews.
In abstract and introduction I have added the objectives of the study as suggested. kindly consider and approve.
In the case report, I have mentioned about the consent that was signed by the patient and have already uploaded the filled consent form.
I have inserted the indications of figures in the text where ever appropriate.
I have added a figure on preoperative CT scan for this case that describes the extent of the disease as written in the text.
As suggested, I have tried to enrich the clinical description of the case with points advised by you.
In discussion the constraints faced by the patient leading to a delayed treatment, have been mentioned.
I have included one conclusion briefing our experience of this case.
I could not address all your queries owing to certain limitations of our setup and patient profile. I indeed tried to input as much as I could. I sincerely regret if the manuscript falls short in some aspects. I would request you to approve and kindly consider it for publication.
Regards.

Round 2

Reviewer 1 Report

Thank you for revisional change made on manuscript. 

Reviewer 2 Report

I congratulate the authors for the excellent reconstruction of the manuscript following the recommendations of these reviewers.